# Elements of cosmology beyond FLRW

**Pierre Fleury**

Laboratoire Univers et Particules de Montpellier (LUPM), CNRS & Université de Montpellier,
Parvis Alexander Grothendieck, F-34095 Montpellier Cedex 05, France

pierre.fleury@lupm.in2p3.fr

## Abstract

Modern cosmology is based on the cosmological principle, which states that the Universe is statistically homogeneous and isotropic. When applied in its strict – rather than statistical – sense, the cosmological principle leads to the Friedmann–Lemaître–Robertson–Walker (FLRW) model, which serves as background spacetime. This background is used to predict: (1) the dynamics of cosmic expansion; and (2) the kinematics of light propagation through the Universe, which dictates the interpretation of cosmological observations. In this lecture, we shall discuss the performance of the FLRW model for those purposes, and present some results on the so-called backreaction and fitting problems.

# 1  Motivation

When proposing, in 1917, the very first cosmological model based on general relativity, Einstein made three simplifying assumptions: staticity, homogeneity and isotropy [1]. While staticity was falsified by the discovery of cosmic expansion in 1929, the premises of homogeneity and isotropy, which we may collectively refer to as the *cosmological principle*, remain part of the foundations of modern cosmology. The later observations of the tiny anisotropies of the cosmic microwave background (CMB), and of the distribution of galaxies on large scales, together with the belief that we do not occupy a special place in the Universe, came to confirm the relevance of the cosmological principle.

Of course we know that the Universe is not strictly homogeneous and isotropic, and the cosmological principle must be understood in a *statistical* sense, when structures are smoothed out on scales of hundreds of megaparsecs. But for simplicity, the standard description of the cosmos is built upon a background spacetime where the cosmological principle is applied in its strict sense, namely the Friedmann–Lemaître–Robertson–Walker (FLRW) model. Most the interesting physics happening in the Universe, such as the dynamics of the primordial plasma, or the formation of structures, is then described as perturbations over the FLRW background.

There are, however, two classes of predictions that directly follow from the FLRW model – without accounting for perturbations. The first one is the dynamics of cosmic expansion; in other words, standard cosmology assumes that inhomogeneities have no impact on how the Universe expands on the largest scales. Assessing the validity of this assumption is known as the *backreaction problem*, which we shall discuss in the first part (section 2) of the lecture. The second class of direct predictions of the FLRW model is the propagation of light through the Universe, and in particular the relation between redshift and distance measures, which are used in the interpretation of all cosmological observations. Whether or not the FLRW is the right framework to perform such an interpretation is known as the *fitting problem*, and will be the topic of the second part (section 3) of the lecture.

# 2  Dynamics: the backreaction hypothesis

## 2.1  General idea of backreaction

The FLRW model describes the dynamics and time evolution of an idealised universe where all inhomogeneities have been smoothed out. In other words, smoothing is performed first, and time evolution follows. Yet, in principle we would rather be interested in the converse procedure, whereby smoothing would be performed after time evolution. This raises the question of the *commutation* between smoothing and time evolution.

As emphasised by Ellis in 1984 [2], smoothing is implicit in cosmological modelling, and defining precisely what a smoothed spacetime means turns out to be quite challenging. A key argument for questioning the commutation of smoothing and dynamics stems from the non-linearity of Einstein's equation. Let $\langle \ldots \rangle$ denote some smoothing or averaging procedure. Since the Einstein tensor $E_{\mu\nu}$ is a non-linear functional of the spacetime metric $g_{\mu\nu}$, the smoothed field equation is generally different from the field equation obeyed by the smoothed metric:

$$E_{\mu\nu}[\langle g_{\rho\sigma} \rangle] \neq \langle E_{\mu\nu}[g_{\rho\sigma}] \rangle = 8\pi G \langle T_{\mu\nu} \rangle. \tag{1}$$

As a consequence, there is a priori no guarantee that the Friedmann equations accurately describe the expansion dynamics of the actual, inhomogeneous Universe.

This is what backreaction is about. The name comes from the idea that, as structures form across cosmic history, the Universe departs more and more from the idealised FLRW model

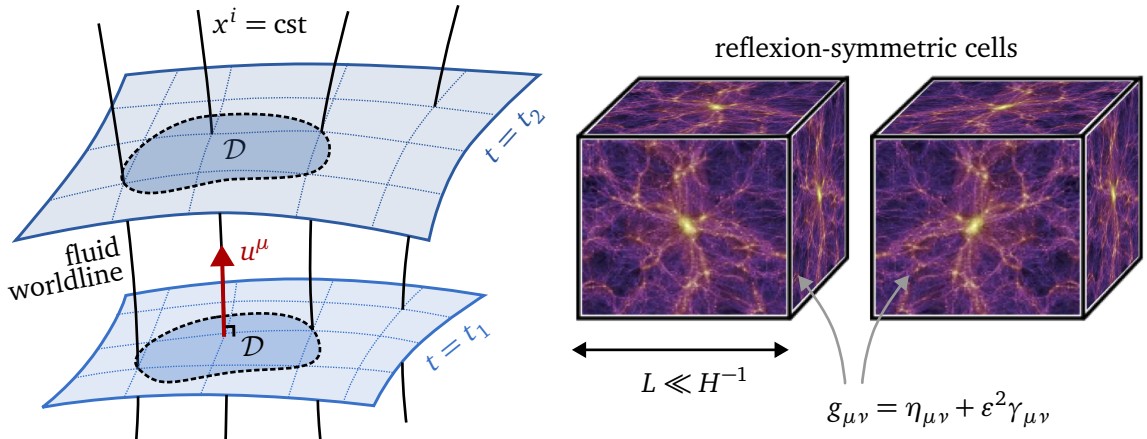

Figure 1: **Left:** geometry of an irrotational congruence of timelike geodesics. **Right:** Post-Newtonian cosmological modelling (adapted from ref. [5]).

and those structures, or inhomogeneities, may backreact on the expansion dynamics itself. The idea got some attention with the discovery of cosmic acceleration in the late 1990s and the *coincidence argument*: is it a coincidence that dark energy starts dominating the expansion dynamics in the late Universe, when it is most inhomogeneous? Could backreaction mimic the effect of dark energy? We shall discuss this point in the present section; for further details, I recommend the short but yet comprehensive review by Clarkson et al. [3].

## 2.2 Buchert's scalar formalism

In the early 2000s, T. Buchert proposed a scalar formalism [4] to describe the average dynamics of an inhomogeneous universe filled with a pressureless fluid (dust). This approach being simple and intuitive, it became quite popular in the subsequent years. Although the resulting backreaction effect is likely small, the method itself remains instructive.

**Description of an inhomogeneous dust flow**  Consider a pressureless fluid; we denote with $u^\mu$ its four-velocity field, which is everywhere tangent to the worldline of the fluid particles. Since the fluid is only subject to gravity, the fluid elements are freely falling, and hence they follow timelike geodesics, $u^\nu \nabla_\nu u^\mu = 0$.

In addition, we assume that the flow is irrotational, $\nabla_\mu u_\nu = \nabla_\nu u_\mu$. This is equivalent to assuming that the family of geodesics representing the fluid's motion is hypersurface-orthogonal: spacetime can be foliated into spatial hypersurfaces that are everywhere orthogonal to $u^\mu$ (see fig. 1, left). Those hypersurfaces physically represent space in the rest frame of an observer following the fluid's motion. As such, they can be labelled by the proper time $t$ along the fluid elements' worldlines. When seen as a field, $t$ satisfies $u_\mu = \partial_\mu t$.

**Comoving–synchronous coordinates**  We may choose $t$ as our time coordinate. As for spatial coordinates $(x^i)_{i \in \{1,2,3\}}$, let them be labels for the fluid elements, so that any $x^i = \mathrm{cst}$ curve is the worldline of some particle in the flow. The resulting coordinate system $(t, x^i)$ is called comoving–synchronous. In terms of such coordinates, the fluid's four-velocity reads $u_\mu = \delta^0_\mu$, and the spacetime line element is

$$\mathrm{d}s^2 = g_{\mu\nu}\mathrm{d}x^\mu\mathrm{d}x^\nu = -\mathrm{d}t^2 + h_{ij}(t, x^k)\,\mathrm{d}x^i\mathrm{d}x^j \,, \tag{2}$$

where $h_{ij}(t, x^k)$ is the spatial metric of the $t = \mathrm{cst}$ hypersurfaces.

This may be seen as the generalisation of cosmic time and comoving coordinates for the standard FLRW geometry. The difference is that, here, the rest-frame density of the fluid is not assumed to be homogeneous, $\rho(t, x^i)$, and consequently there is in general no choice for $(x^i)$ such that $h_{ij}$ is a function of time only.

**Volume of a comoving region**    We aim to determine the global expansion dynamics of this inhomogeneous fluid. We are thus interested in the time evolution of the physical volume of some very large comoving spatial domain $\mathcal{D}$,

$$V_{\mathcal{D}}(t) = \int_{\mathcal{D}} \mathrm{d}^3 x \ \sqrt{h} \,, \qquad h \equiv \det[h_{ij}]. \tag{3}$$

Since the coordinate system is following the fluid's motion, the latter is entirely encoded in the spatial metric $h_{ij}$.[1] The evolution of $V_{\mathcal{D}}$ then depends on the average dynamics of $h$.

**Raychaudhuri equation**    The local dynamics of $h$ directly follows from the geometry of geodesic congruences. We first define the *local expansion rate* $\theta$ from the time-evolution of a very small region,

$$\theta(t, x^i) = \frac{1}{\delta V} \frac{\mathrm{d}\delta V}{\mathrm{d}t} = \frac{1}{\sqrt{h}} \frac{\partial \sqrt{h}}{\partial t} = \frac{1}{2} h^{ij} h_{ij,t} \,, \tag{4}$$

which may be seen as a local generalisation of $3 \times H$ in an inhomogeneous context.

It is straightforward to check that $\theta = \nabla_\mu u^\mu = \nabla_i u^i$. More generally, since $u^\nu \nabla_\nu u^\mu = 0$ the symmetric tensor $\nabla_\mu u_\nu$ is purely spatial; we may decompose it into a pure-trace and a trace-free part as

$$\nabla_i u_j = \frac{1}{3} \theta h_{ij} + \sigma_{ij} \,, \tag{5}$$

where $\sigma_{ij}$ is the shear tensor of the flow, which describes its rate of deformation with time. Taking the time derivative of the expansion rate, $\mathrm{d}\theta/\mathrm{d}t = u^\nu \nabla_\nu (\nabla_\mu u^\mu) = -\nabla_\mu u_\nu \nabla^\mu u^\nu - R_{\mu\nu} u^\mu u^\nu$ and substituting Einstein's equation then yields the Raychaudhuri equation:

$$\frac{\mathrm{d}\theta}{\mathrm{d}t} = -\frac{1}{3}\theta^2 - 2\sigma^2 - 4\pi G\rho + \Lambda \,, \qquad \sigma^2 \equiv \frac{1}{2}\sigma^{ij}\sigma_{ij} \,. \tag{6}$$

Equation (6) is reminiscent of the second Friedmann equation, and shows that the local expansion dynamics results from a competition between the cosmological constant, which tends to accelerate it, and the matter density and shear, which tend to slow it down.

**Global expansion dynamics: backreaction**    Let us now examine the consequences of eq. (6) on the global expansion dynamics of $\mathcal{D}$. We first define an effective scale factor $a_{\mathcal{D}}$ as

$$V_{\mathcal{D}}(t) = a_{\mathcal{D}}^3(t) V_{\mathcal{D}}(t_0) \,. \tag{7}$$

Then, taking the second derivative of $a_{\mathcal{D}}$, with some algebra and the use of the volume-average of the Raychaudhuri equation (6), we find the following effective Friedmann equation for $\mathcal{D}$,

$$\boxed{\frac{1}{a_{\mathcal{D}}} \frac{\mathrm{d}^2 a_{\mathcal{D}}}{\mathrm{d}t^2} = -\frac{4\pi G}{3}\langle\rho\rangle_{\mathcal{D}} + \frac{\Lambda}{3} + \underbrace{\frac{1}{3}\left[\frac{2}{3}\left(\langle\theta^2\rangle_{\mathcal{D}} - \langle\theta\rangle_{\mathcal{D}}^2\right) - 2\langle\sigma^2\rangle_{\mathcal{D}}\right]}_{\text{backreaction term } \mathcal{B}}}, \tag{8}$$

---

[1]This does not mean that "space expands", as often written in the public-outreach literature and sometimes, unfortunately, in cosmology textbooks [6]. Cosmic expansion is about matter, not about space which has no substance, especially in relativity where it is coordinate-dependent. The interpretation of an expanding space directly comes from the use of comoving coordinates. See [7] for a diatribe on the expanding space.

with the volume-average operator

$$\langle X \rangle_{\mathcal{D}} \equiv \frac{1}{V_{\mathcal{D}}} \int_{D} \mathrm{d}^3 x \ \sqrt{h}\, X \ . \tag{9}$$

Compared to the usual Friedmann equation, eq. (8) exhibits an additional, *backreaction term* $\mathcal{B}$, which could mimic the effect of dark energy [8].

**Caveats**   Buchert's spatial averaging formalism can only deal with scalar quantities. As such, part of the microscopic dynamics is lost in the process, and the resulting set of dynamical equations for $\mathcal{D}$ is not closed, which makes it hard to quantify the backreaction effect.

The backreaction term in eq. (8) results from the competition between the variance of the local expansion rate, $\langle \theta^2 \rangle_{\mathcal{D}} - \langle \theta \rangle_{\mathcal{D}}^2$, and the mean square of the shear rate, $\langle \sigma^2 \rangle_{\mathcal{D}}$. It is quite easy to see that there is more shear if the expansion rate is more inhomogeneous, so that both terms tend to cancel out. As it turns out, in Newtonian cosmology $\mathcal{B}$ is a boundary term over $\partial \mathcal{D}$, divided by the volume $V_{\mathcal{D}}$ [9]; hence it is negligible for large domains $\mathcal{D}$. This suggests that cosmological backreaction, if anything, must come from relativistic effects.

## 2.3   Relativistic backreaction

Other authors chose to focus more specifically the general-relativistic origin of backreaction. Here is a selection of three such approaches, which all suggest a negligibly small effect.

**Clifton & Sanghai's post-Newtonian patchwork**   This is a construction based on the post-Newtonian (PN) expansion of general relativity [5]. In this approach, the spacetime metric is determined in a cubic cell $\mathcal{C}$ with size $L \ll H^{-1}$ using the PN expansion $g_{\mu\nu} = \eta_{\mu\nu} + \varepsilon^2 \gamma_{\mu\nu}$, where $\varepsilon \sim v/c$ is the PN expansion parameter, $v$ being the typical velocity of matter.

Carefully patching together identical copies of $\mathcal{C}$ results in a lattice universe (see fig. 1, right panel), whose expansion dynamics emerges from the small-scale physics. In this construction, there appears a genuinely relativistic backreaction term [10]

$$\mathcal{B} \approx \frac{2}{3} \frac{(\Omega_{\mathrm{m}} H_0)^2}{a^4} (H_0 L_0)^2 \sim \varepsilon^4 \ , \tag{10}$$

where $L_0$ is the present length of $\mathcal{C}$ and the scale factor is identified as $a(t) = L(t)/L_0$. This term contributes positively to the acceleration of cosmic expansion and dilutes like radiation ($\propto a^{-4}$). Because it is suppressed by $(H_0 L_0)^2 \ll 1$ compared to the other terms in the Friedmann equation, its effect is negligible in practice.

**Green & Wald's effective field theory**   In ref. [11], the spacetime metric is assumed to read $g_{\mu\nu} = \bar{g}_{\mu\nu} + \gamma_{\mu\nu}$, where $\bar{g}_{\mu\nu}$ is a smooth effective metric (e.g. FLRW), and $\gamma_{\mu\nu}$ encodes the small-scale physics. While $\gamma_{\mu\nu}$ is assumed to be small, its first derivatives $\nabla_{\rho} \gamma_{\mu\nu}$ are only required to remain finite, and its second derivatives $\nabla_{\rho} \nabla_{\sigma} \gamma_{\mu\nu}$ may be arbitrarily large.

Under these assumptions, the authors of ref. [11] demonstrated that $\bar{g}_{\mu\nu}$ must satisfy an effective Einstein equation, where the backreaction due to $\gamma_{\mu\nu}$ manifests a correction $t_{\mu\nu}$ to the stress-energy tensor, with $t_\mu^\mu = 0$. This extra contribution is interpreted as the small-scale gravitational radiation that is integrated out in the dynamics of $\bar{g}_{\mu\nu}$, thereby producing a negligible backreaction effect in cosmology [12]. Such conclusions led to a heated debate within the backreaction community [13, 14].

149  GEVOLUTION   In 2016, the Geneva cosmology group addressed the backreaction problem
150  from a numerical perspective. GEVOLUTION [15] is a relativistic $N$-body code in which the
151  space-time metric reads $g_{\mu\nu} = \bar{g}_{\mu\nu} + \gamma_{\mu\nu}$, where the background $\bar{g}_{\mu\nu}(t)$ is formally similar to
152  the FLRW metric, but contrary to other cosmological simulations, it does not pre-suppose its
153  evolution based on the Friedmann equations; instead, the definition of the scale factor is up-
154  dated at each time step. Computations are then performed at first order in $\gamma_{\mu\nu}$, second order
155  in $\nabla_\rho \gamma_{\mu\nu}$, and arbitrary order in $\nabla_\rho \nabla_\sigma \gamma_{\mu\nu}$. The expansion law is then found to match the
156  usual Friedmannian dynamics up to a part in $10^4$ [16].

# 3   Observations: the fitting problem

158  Even if the backreaction of structures on cosmic expansion is negligible, this does not neces-
159  sarily imply that the FLRW model is the right framework to accurately interpret cosmological
160  observations, because we do not directly observe physical distances and velocities, but rather
161  redshifts, angles and apparent brightnesses.
162      In this section, we shall focus on the relation between redshift and angular (or luminosity)
163  distance, which is ubiquitous in the interpretation of cosmological data, from the Hubble di-
164  agram of type-Ia supernovae (SNe) to the CMB anisotropies, including the imprint of baryon
165  acoustic oscillations (BAO) in the distribution of galaxies.

## 3.1   The distance–redshift relation

167  **Definitions**   The redshift is the change of a signal's cyclic frequency in the observer's frame
168  relative to the source's frame, $1 + z \equiv \omega_s/\omega_o$. The angular-diameter distance, or area dis-
169  tance $D_A$ is defined so as to translate the observed angular size $\Omega_o$ of an image (which is a
170  solid angle) to the physical area $A_s$ of its source,

$$A_s = D_A^2 \times \Omega_o \,. \tag{11}$$

171  The luminosity distance $D_L$, on the other hand, connects the observed flux $F_o$ (power per unit
172  area) of a light source to is intrinsic luminosity (power) $L_s$ as

$$L_s = F_o \times 4\pi D_L^2 \,. \tag{12}$$

173  Both notions of distance are connected by the distance-duality relation, $D_L = (1 + z)^2 D_A$,
174  which is valid if light follows null geodesics, and if there is no significant photon creation or
175  absorption during light propagation from the source to the observer. We may thus focus on
176  the angular-diameter distance from now on.

177  **Relation in FLRW**   When light is emitted and observed by comoving entities in a strictly
178  homogeneous and isotropic universe, the distance–redshift relation is found to read

$$\bar{D}_A(z) = \frac{1}{1+z} \frac{\sinh[\sqrt{-K}\chi(z)]}{\sqrt{-K}} \,, \qquad \chi(z) = \int_0^z \frac{dz'}{H(z')} \,, \tag{13}$$

179  where $K$ denotes the curvature of the homogeneity hypersurfaces ("space"), and $\chi$ is known
180  as the comoving distance. Barred quantities will refer to FLRW in the whole section. Equa-
181  tion (13) is used to constrain cosmological parameters from the Hubble diagram of type-Ia
182  SNe, but also from BAO and CMB observations.

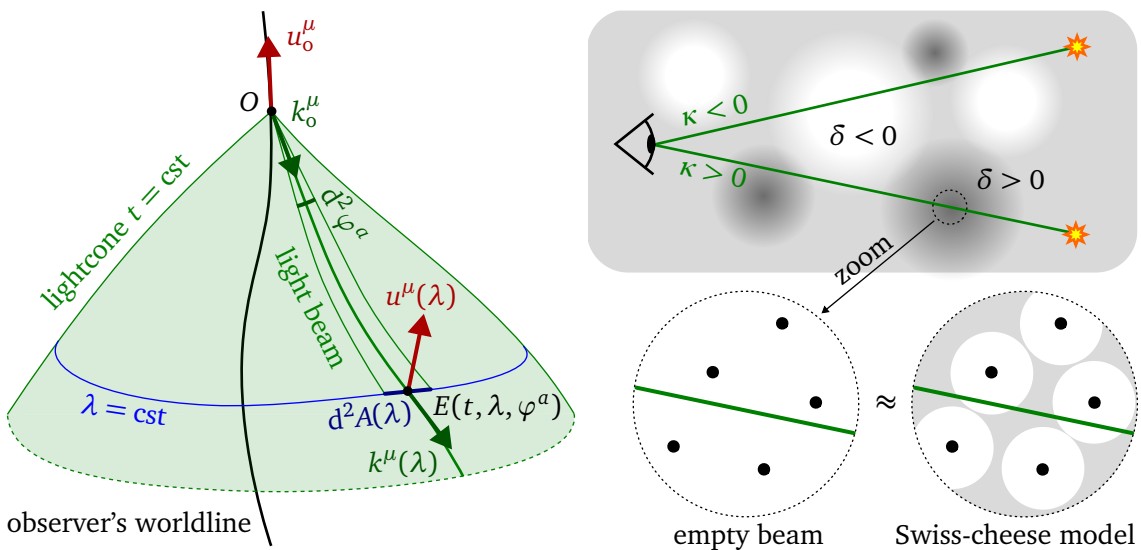

Figure 2: **Left**: Geometrical construction of the observational coordinates $(t, \lambda, \varphi^a)$. **Right**: Overdense (underdense) lines of sight have positive (negative) convergence $\kappa$; in linear perturbation theory, one expect $\langle \kappa \rangle = 0$. However, accounting for the small-scale lumpiness of the matter distribution suggests instead an empty-beam model, corresponding to light propagation in, e.g., a Swiss-cheese universe.

**In general, the distance–redshift relation is twofold** The standard result (13) hides the fact that $D_A(z)$ is fundamentally the combination of two relations: $z(\lambda)$ and $D_A(\lambda)$, where $\lambda$ is an affine parameter along the ray, i.e. the null geodesic, connecting the source to the observer.

Let $k^\mu$ be the past-directed wave four-vector of that ray, that is a tangent vector to the null geodesic (see fig. 2, left). The cyclic frequency measured by an observer with four-velocity $u^\mu$ is $\omega = u_\mu k^\mu$, hence $1 + z(\lambda) = (u_\mu k^\mu)(\lambda)$ if we conventionally set $\omega_o = 1$. Taking the derivative with respect to $\lambda$ then yields

$$\frac{dz}{d\lambda} = k^i k^j \nabla_i u_j \equiv \frac{1}{3}(1 + z)^2 \theta_\| \,, \tag{14}$$

where $\theta_\|$ represents the local expansion rate of the matter fluid along the spatial direction of light propagation $\hat{\boldsymbol{k}}$. If backreaction is small, then $\theta_\| \approx 3H$ on average, which implies that $z(\lambda)$ is mostly unaffected by inhomogeneities.

Regarding the second part of the relation, $D_A(\lambda)$, things are somewhat more complicated because they depend on the evolution, with light propagation, of the physical area $d^2 A(\lambda)$ of a small light beam subtended by the angle $d^2 \Omega_o$ at the observer.

## 3.2 Propagation of light beams in general relativity

**Observational coordinates** The general study of light beams is more convenient as one writes the metric in terms of a coordinate system adapted to null geodesics [17]. Those observational coordinates $(t, \lambda, \varphi^1, \varphi^2)$ are defined relative to a fiducial observer (see fig. 2, left).

Let $E$ be an event in spacetime, which we may see as the emission of a photon received at $O$ by the observer. The first coordinate of $E$ is taken to be the proper time $t$ of $O$ along the observer's worldline. Thus, $t = $ cst is the past lightcone of $O$. We may choose the second coordinate, $\lambda$, as the affine parameter of $E$ along the null geodesic connecting it to $O$. The surfaces of constant $(t, \lambda)$ can be seen as wavefronts converging to $O$. The last two coordinates, $(\varphi^1, \varphi^2)$, indicate the direction in which a photon emitted at $E$ is observed at $O$ in the observer's

206 frame. Curves with constant $(t, \varphi^1, \varphi^2)$ thus represent light rays converging to $O$. In that sense,
207 $(\varphi^1, \varphi^2)$ may be seen as comoving coordinates for photons.

208    One can show that any spacetime metric in observational coordinates reads

$$\mathrm{d}s^2 = g_{\mu\nu}\mathrm{d}x^\mu\mathrm{d}x^\nu = -N\mathrm{d}t^2 - \mathrm{d}t\mathrm{d}\lambda + k_a\mathrm{d}t\mathrm{d}\varphi^a + S_{ab}\mathrm{d}\varphi^a\mathrm{d}\varphi^b \,, \tag{15}$$

209 with $a, b \in \{1, 2\}$, and $N, k_a, S_{ab}$ are six free functions of $(t, \lambda, \varphi^a)$. From now on, we shall work
210 exclusively on the lightcone of $O$, hence $\mathrm{d}t = 0$, and the metric reduces to $\mathrm{d}s^2 = S_{ab}\mathrm{d}\varphi^a\mathrm{d}\varphi^b$.

**Angular-diameter distance**   Equipped with the observational metric (15), we can conve-
212 niently express the cross-sectional area of a light beam, which is the portion of wavefront
213 corresponding to some angular region $\mathcal{I}$ (an image) on the observer's celestial sphere,

$$A(\lambda) = \int_{\mathcal{I}} \mathrm{d}^2\varphi^a \sqrt{S} \,, \qquad S \equiv \det[S_{ab}]. \tag{16}$$

214 With the right choice of $\varphi^a$, we have $\mathrm{d}^2\varphi^a = \mathrm{d}^2\Omega_\mathrm{o}$ and hence

$$D_\mathrm{A}^2 = \sqrt{S} \,. \tag{17}$$

**Focusing theorem**   Similarly to the case of timelike geodesics seen in section 2, we can define
216 the local rate of expansion of the wavefront with $\lambda$,

$$\Theta \equiv \frac{1}{\delta A}\frac{\mathrm{d}\delta A}{\mathrm{d}\lambda} = \frac{1}{\sqrt{S}}\frac{\partial\sqrt{S}}{\partial\lambda} = \frac{1}{2}S^{ab}S_{ab,\lambda} \,. \tag{18}$$

217 It is straightforward to check that $\Theta = \nabla_\mu k^\mu = \nabla_a k^a$; more generally, we may decompose
218 the tensor $\nabla_a k_b$ into a pure-trace and a trace-free part as $\nabla_a k_b = (1/2)\Theta S_{ab} + \Sigma_{ab}$, where
219 $\Sigma_{ab}$ is the shear rate of the light beam. Physically speaking, while $\Theta$ is mostly sourced by the
220 local density intercepted by the light beam, $\Sigma_{ab}$ is sourced by the tidal forces produced by
221 concentrations of matter near the beam.

222    Let us analyse the evolution of $\Theta$ down the observer's lightcone. Differentiating $\Theta$ with
223 respect to $\lambda$, and using $k^\nu\nabla_\nu k^\mu = 0$, we get $\mathrm{d}\Theta/\mathrm{d}\lambda = k^\nu\nabla_\nu(\nabla_\mu k^\mu) = -\nabla_\mu k_\nu\nabla^\mu k^\nu - R_{\mu\nu}k^\mu k^\nu$.
224 Then, substituting the decomposition of $\nabla_a k_b$ into expansion and shear rates, together with
225 Einstein's equation, yields the *null Raychaudhuri equation*:

$$\frac{\mathrm{d}\Theta}{\mathrm{d}\lambda} = -\frac{1}{2}\Theta^2 - 2\Sigma^2 - 8\pi G(1+z)^2\rho \,, \qquad 2\Sigma^2 = \Sigma^{ab}\Sigma_{ab} \,. \tag{19}$$

226    We may further substitute $\Theta = (2/D_\mathrm{A})\mathrm{d}D_\mathrm{A}/\mathrm{d}\lambda$ and get the *focusing theorem*

$$\boxed{\frac{1}{D_\mathrm{A}}\frac{\mathrm{d}^2 D_\mathrm{A}}{\mathrm{d}\lambda^2} = -4\pi G(1+z)^2\rho - \Sigma^2 \leq 0.} \tag{20}$$

227 Equation (20) implies that $\mathrm{d}^2 D_\mathrm{A}/\mathrm{d}\lambda^2 \leq 0$, so light beams cannot be defocused.[2]   As light
228 propagates through some smooth matter distribution, focusing is ensured by the first term
229 $\propto \rho$; when it passes near mass lumps, focusing occurs through the shear rate $\Sigma^2$ produced by
230 tidal forces. One may also note that $\Lambda$ is absent from eq. (20): while the cosmological constant
231 pushes massive particles away, it has no effect on photons.

---

[2]A riddle for strong lensers: given eq. (20), how can the secondary image of, e.g., a point lens, be defocused?

## 3.3  Optics in a lumpy Universe

In the FLRW model, $\rho = \bar{\rho} = (1+z)^3 \bar{\rho}_0$ and $\Sigma = 0$. Let us now examine the consequences of eq. (20) on optics in an inhomogeneous Universe.

**Linear perturbation theory**  As a warm-up exercise, we may consider a perturbed FLRW model with $\rho = \bar{\rho}(1+\delta)$, where $\delta$ denotes the density contrast. At first order in $\delta$, we may neglect the shear term $\Sigma^2 = \mathcal{O}(\delta^2)$. Solving eq. (20) then yields the following correction relative to the FLRW distance,

$$\frac{\delta D_A(z, \varphi^a)}{\bar{D}_A(z)} = -4\pi G \bar{\rho}_0 \int_0^{\chi(z)} \mathrm{d}\chi' \, \frac{\chi'[\chi(z) - \chi']}{\chi(z)} \frac{\delta(\eta_0 - \chi', \chi', \varphi^a)}{a(\eta_0 - \chi')} \equiv -\kappa(z, \varphi^a) \, . \qquad (21)$$

where we assumed a spatially flat background ($K = 0$) for simplicity. In the above, $\eta_0$ denotes the conformal time today. This perturbative correction is known as the weak-lensing convergence $\kappa$. It is positive when the line of sight is mostly overdense and negative when it is mostly underdense (see fig. 2, top-right panel). In this framework, however, it is easy to see that the correction vanishes when averaged over the observer's sky, $\langle \kappa \rangle = 0$, which suggests that inhomogeneities cause no systematic bias in the distance–redshift relation.

**Zel'dovich's empty beam**  An argument against the above conclusion was put forward in 1964 by Zel'dovich [18]. At the scale of the tiny light beams relevant to, e.g., SN observations, the Universe is extremely lumpy, so that such light beams should mostly propagate through vacuum ($\rho \approx 0$), even in overdense regions of the Universe (fig. 2, bottom panel). If furthermore the matter lumps making up the mean cosmic are opaque and not too compact, then shear may be neglected, $\Sigma^2 \approx 0$, which leads to the empty-beam model

$$D_A^{\mathrm{EB}}(z) = \lambda(z) = \int_0^z \frac{\mathrm{d}z'}{(1+z')^2 H(z')} > \bar{D}_A(z) \, . \qquad (22)$$

The relative difference between $D_A^{\mathrm{EB}}(z)$ and $\bar{D}_A(z)$ can be significant. In a flat $\Lambda$CDM cosmology, it reaches about 10% for $z = 1$ [19].

**Swiss-cheese models**  Zel'dovich's intuition can be made more rigorous by considering the so-called Swiss-cheese model [20] (fig. 2, bottom-right panel). The construction is the following; starting from the FLRW model, pick a sphere with comoving radius $\chi$ and concentrate the matter that it contains by a point mass $M = (4\pi/3)\bar{\rho}(a\chi)^3$, thereby making a "hole" in the otherwise homogeneous "cheese". Spacetime inside the hole is described by the Schwarzschild geometry, which turns out to glue perfectly with FLRW at the boundary of the hole. An arbitrary amount of such holes can then be introduced as long as they do not overlap.

Light propagation in Swiss-cheese models was first investigated by Kantowski [21] and later by Dyer & Roeder [22], who concluded that light beams effectively propagate in an underdense universe with negligible shear $\Sigma^2$,

$$\frac{1}{D_A^{\mathrm{KDR}}} \frac{\mathrm{d}^2 D_A^{\mathrm{KDR}}}{\mathrm{d}\lambda^2} = -4\pi G(1+z)^2 \alpha \rho \, , \qquad (23)$$

where $\alpha \equiv V_{\mathrm{FLRW}}/V \leq 1$ is the smoothness parameter of the model. The KDR model (23) thus interpolates between the empty-beam model ($\alpha = 0$) and the FLRW prediction ($\alpha = 1$). It has been shown numerically to provide a good approximation to Swiss-cheese models [19, 23] as well as other lumpy models [24], including the PN patchwork model mentioned in section 2.3 if the opacity radius $R$ of the matter lumps is large enough.

**The crucial role of shear**   As the opacity radius $R$ decreases, that is, as light beams are allowed to pass closer to the matter lumps, the contribution $\Sigma^2$ of shear increases. Using a stochastic model, ref. [25] quantified the contribution of shear to the angular distance in a Swiss-cheese model; at $z \approx 1$ this correction to the KDR distance scales as

$$\left( \frac{\delta D_{\mathrm{A}}}{D_{\mathrm{A}}^{\mathrm{KDR}}} \right)_{\Sigma^2} \approx -10^{-3} \mathcal{A} \,, \qquad \text{with} \quad \mathcal{A} \equiv \frac{2GM}{H_0 R^2} \,, \tag{24}$$

for opaque lumps of mass $M$ and radius $R$. If the matter lumps represent galaxies, then $\mathcal{A} \sim 1$, but if they represent individual stars, then $\mathcal{A} \sim 10^{10}$ so that the shear contribution blows up. This is a hint that $\Sigma^2$ should actually be able to compensate the deficit of focusing due to the smaller effective matter density, when properly taken into account.

In fact, it was shown by Weinberg [26] that the *average* of $D_{\mathrm{A}}(\lambda)$ over directions in a universe filled with point masses is identical to the $\bar{D}_{\mathrm{A}}(\lambda)$ of a universe homogeneously filled with matter with the same density. Although the calculation was performed at first order in the mean matter surface density (or optical depth), this indicates that the shear term $\Sigma^2$ in eq. (20) can indeed cancel out the effect of local density fluctuations.

## 3.4   Observational averages

**Magnification theorems**   The aforementioned result from Weinberg can be generalised as follows. Consider an infinitesimal light source with apparent size $\mathrm{d}^2\Omega_{\mathrm{s}}$ if it were observed through an FLRW universe (unlensed size); the magnification of that source is defined as

$$\mu \equiv \frac{\mathrm{d}^2\Omega_{\mathrm{o}}}{\mathrm{d}^2\Omega_{\mathrm{s}}} = \left[ \frac{\bar{D}_{\mathrm{A}}}{D_{\mathrm{A}}} \right]^2 \,, \tag{25}$$

where $\mathrm{d}^2\Omega_{\mathrm{o}}$ is the observed angular size of the image as observed through the actual, inhomogeneous, Universe. A source is magnified ($\mu > 1$) if the corresponding light beam propagates through overdense regions of the Universe ($\rho > \bar{\rho}$), or regions with a high shear rate $\Sigma$.

Suppose now that such sources are homogeneously covering the celestial sphere $\mathcal{S}^2$, the mean magnification of such a set of sources then reads

$$\langle \mu \rangle_{\mathrm{s}} = \frac{1}{4\pi} \int_{\mathcal{S}^2} \mathrm{d}^2\Omega_{\mathrm{s}} \, \mu = \frac{1}{4\pi} \int_{\mathcal{S}^2} \mathrm{d}^2\Omega_{\mathrm{o}} = 1 \,, \tag{26}$$

where $\langle \cdots \rangle_{\mathrm{s}}$ stands for *source averaging*, because it gives the same weight to sources with the same intrinsic size $\mathrm{d}^2\Omega_{\mathrm{s}}$. If, on the contrary, we choose to give the same weight to all directions in the sky, then it is the inverse magnification that averages to unity:

$$\langle \mu^{-1} \rangle_{\mathrm{o}} = \frac{1}{4\pi} \int_{\mathcal{S}^2} \mathrm{d}^2\Omega_{\mathrm{o}} \, \mu^{-1} = \frac{1}{4\pi} \int_{\mathcal{S}^2} \mathrm{d}^2\Omega_{\mathrm{s}} = 1 \,, \tag{27}$$

where $\langle \cdots \rangle_{\mathrm{o}}$ stands for *directional averaging*. The magnification theorems (26) and (27) are trivial when all light sources have a single image, but they also hold in the presence of multiple imaging – see e.g. [27] and references therein.

**Bias on distances**   Directional averaging $\langle X \rangle_{\mathrm{o}}$ may be understood as what an observer would measure by observing $X$ in random directions of the sky; it is the most natural notion of average involved in numerical ray-tracing simulations. Source-averaging $\langle X \rangle_{\mathrm{s}}$, on the other hand, is generally more relevant to actual observations. As such, eq. (26) can be used to derive the observational bias on the observed distance to SNe due to inhomogeneities in the Universe:

$$\langle D_{\mathrm{A}}(z) \rangle_{\mathrm{s}} = \bar{D}_{\mathrm{A}}(z) \langle \mu^{-1/2} \rangle_{\mathrm{s}} = \bar{D}_{\mathrm{A}}(z) \left[ 1 + \frac{3}{2} \langle \kappa^2(z) \rangle + \dots \right], \tag{28}$$

where $\kappa$ is the weak-lensing convergence defined in eq. (21). Note that the bias is of second order in cosmological perturbations; this is why $\langle \kappa^2 \rangle$ can be either a source average or a directional average since the difference would be of higher order. Using Limber's approximation, one shows that [28, 29]

$$\langle \kappa^2(z) \rangle = \frac{1}{4\pi} \sum_{\ell=0}^{\infty} (2\ell + 1)\, C_\ell^\kappa(z) \,, \tag{29}$$

$$C_\ell^\kappa(z) \approx (4\pi G \bar{\rho}_0)^2 \int_0^{\chi(z)} \mathrm{d}\chi' \left[ \frac{\chi'(\chi - \chi')}{a(\eta_0 - \chi')\chi} \right]^2 P_{\mathrm{m}} \left[ \eta_0 - \chi', \frac{\ell + 1/2}{\chi'} \right], \tag{30}$$

where $P_{\mathrm{m}}(\eta, k)$ denotes the matter power spectrum. In a $\Lambda$CDM cosmology, at $z = 1$ the above implies that the bias on the average distance (28) reaches about $10^{-3}$, which is negligible in practice. However, for the CMB at $z_* = 1090$, the bias reaches 5%, which is significant.

**The distance-to-the-CMB controversy**  Does this mean, as argued in ref. [30], that the standard CMB analysis, based on the FLRW expression for the angular-diameter distance, is flawed? Fortunately, it does not, for three reasons. First of all, CMB analyses do not directly depend on $\langle D_{\mathrm{A}}(z_*) \rangle_{\mathrm{s}}$, but on the angular power spectrum, which itself relies on directional averaging $\langle \cdots \rangle_{\mathrm{o}}$ rather than source-averaging [31].

Second, and most importantly, the expression (29) of $\langle \kappa^2 \rangle$ implicitly accounts for cosmological perturbations down to arbitrarily small scales ($\ell \to \infty$). This would be correct if the characteristic angular scales measured in the CMB were infinitesimal; but in reality the angular-diameter distance to the CMB is used to convert the angular size of the sound horizon $\theta_* \approx 0.6\,\mathrm{deg}$ into a physical distance $r_{\mathrm{s}} = D_{\mathrm{A}}(z_*)\theta_*$. This suggests that, as far as cosmological analyses are concerned, the relevant light beams in the CMB have an aperture $\sim \theta_*$. Now, finite light beams propagating through some matter distribution effectively smooth out inhomogeneities that are smaller than the beam's size [32]; hence the sum over $\ell$ in eq. (29) should actually be cut above $\ell_* = \pi/\theta_* \approx 300$, which yields a bias on $D_{\mathrm{A}}(z_*)$ of about $10^{-3}$.

Third and finally, the impact of gravitational lensing on the interpretation CMB power spectrum at second order is already taken into account in CMB-lensing analyses [33].

# 4   Conclusion and outlook

The cosmological principle, which proposes that the Universe is statistically homogeneous and isotropic, is one of the fundamental premises of modern cosmology. The FLRW model is the simplest application of that principle, and yet it turns out to provide a very efficient basis to cosmological modelling. On the one hand, the *backreaction* of inhomogeneities on the expansion dynamics seems to be practically negligible in the framework of general relativity, although alternative theories of gravitation may lead to a different conclusions [34, 35]. On the other hand, inhomogeneities have a negligible impact on the *average* relation between redshift and distances, which is key to interpreting most cosmological observations, although significant corrections to the FLRW prediction can occur on individual measurements. Summarising, *if the cosmological principle is correct*, then a model built upon the FLRW background should provide an accurate description of the Universe. This conclusion highlights the necessity to test the cosmological principle itself with great precision.

There are, in my opinion, two recent observations whose results are at least intriguing in that respect. In the first one [36], the sky is divided in large pixels were the CMB anisotropies are fitted as if they were all independent; this analysis reveals three large sky patches where the cosmological parameters consistently differ by up to 30% from each other, as if we were

lying at the cross-roads of three distinct universes. The second intriguing observation is based on the *dipole test* proposed by Ellis & Baldwin in 1984 [37]. In a nutshell, if the Universe satisfies the cosmological principle, then the observed dipole in the CMB must be of kinematic origin. Because of aberration effects and kinematic redshift, our velocity with respect to the homogeneity frame also manifests itself in the apparent number density of distance sources, such as radio galaxies or quasars. Reference [38] found that this quasar dipole is twice larger than what is expected from the observed CMB dipole. To my knowledge, there is currently no explanation to those anomalies.

# Acknowledgements

I warmly thank the organisers of the summer school *Dark Universe* at Les Houches for the opportunity to give this lecture, and for providing such a fantastic environment. I also thank all the students for great scientific discussions, stunning hikes, and legendary foosball games. No artificial intelligence was involved in the design and writing of those lecture notes.

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
