# Peer review of "Elements of cosmology beyond FLRW"

_SciPost Physics Lecture Notes_

## Round 1 · Referee Report · Anonymous (Referee 1) · 2025-9-24

Strengths

1- The paper provides a clear overview about cosmological backreaction and the fitting problem, detailing what these topics really mean; 2- It helps fix the main ideas for an audience that has some experience with the topics being treated.

Report

I think that this paper is a valuable overview of two interesting topics in modern Cosmology. In particular, it provides a pedagogical introduction to the problem of backreaction and it addresses how to compute it in a general framework.

Requested changes

1- To improve the quality of the manuscript, I would recommend extending the bibliography, to help non-experts identify further relevant literature. 2- In particular, in Section 3.2, in which light-cone coordinates are introduced, it would be worth mentioning that there are other definitions of light-cone coordinates (e.g. geodesic light-cone coordinates), as another possible approach to compute cosmological observables.

Recommendation

Ask for minor revision

---

## Round 1 · Referee Report · Anonymous (Referee 2) · 2025-10-16

Strengths

1) The notes provide a very clear and transparent outline of the cosmological principle, with a specific focus on its possible limitations and pitfalls.

2) The content is highly relevant in the current era of cosmology, where several datasets hint at a potential breakdown of the cosmological principle. This set of notes serves as an excellent starting point to understand why (if the data are indeed correct) it is important to study the theoretical, physical, and philosophical implications of inhomogeneities.

3) The development of ideas is extremely approachable: starting from the very basics and stopping just before entering specialized scenarios. Ample and appropriate references are provided to guide readers toward further reading.

4) The figures are crisp and well chosen, greatly helping to convey the main ideas of the notes.

Weaknesses

N/A

Report

This work clearly meets the journal’s acceptance criteria. Given the upcoming tensions that the cosmological principle faces in light of observational data, this set of notes provides an ideal introduction to the theoretical implications of a possible breakdown of homogeneity and isotropy.

While the FLRW framework remains, as outlined, highly robust against current data, the notes appropriately emphasize the need for caution in alternative cosmological models. In particular, they highlight the importance of re-examining backreaction and light propagation effects from first principles, especially in exotic theories and non-trivial beyond-FLRW scenarios.

Requested changes

Nothing major, except for a few minor typographical and stylistic suggestions:

1) Lines 30–31: Inserting "of" would make the sentence read more smoothly.

2) Consider replacing "cst" with "const" or "const.".

3) There are minor inconsistencies in punctuation before and after equations. Please recheck for uniformity.

4) Line 126 does not read well grammatically and may benefit from rephrasing.

5) Line 215: Consider using "Similar" instead of "Similarly".

6) Line 353: Consider using "these" instead of "those".

Recommendation

Publish (easily meets expectations and criteria for this Journal; among top 50%)

---

## Round 2 · List of Changes

• corrected minor typos
  • added footnote 2 on GLC coordinates and some references

---

## Editorial Decision

accepted_in_target_journal